# Comparative Transcriptome Analysis Reveals Novel Candidate Resistance Genes Involved in Defence against *Phytophthora cactorum* in Strawberry

**DOI:** 10.3390/ijms241310851

**Published:** 2023-06-29

**Authors:** Anupam Gogoi, Erik Lysøe, Håvard Eikemo, Arne Stensvand, Jahn Davik, May Bente Brurberg

**Affiliations:** 1Department of Plant Sciences, Faculty of Biosciences (BIOVIT), Norwegian University of Life Sciences (NMBU), 1433 Ås, Norway; anupam.gogoi@nibio.no (A.G.);; 2Division of Biotechnology and Plant Health, Norwegian Institute of Bioeconomy Research (NIBIO), 1433 Ås, Norway

**Keywords:** crown rot, disease resistance (*R*-genes), *Fragaria vesca*, oomycete

## Abstract

Crown rot, caused by *Phytophthora cactorum*, is a devastating disease of strawberry. While most commercial octoploid strawberry cultivars (*Fragaria* × *ananassa* Duch) are generally susceptible, the diploid species *Fragaria vesca* is a potential source of resistance genes to *P. cactorum*. We previously reported several *F. vesca* genotypes with varying degrees of resistance to *P. cactorum*. To gain insights into the strawberry defence mechanisms, comparative transcriptome profiles of two resistant genotypes (NCGR1603 and Bukammen) and a susceptible genotype (NCGR1218) of *F. vesca* were analysed by RNA-Seq after wounding and subsequent inoculation with *P. cactorum.* Differential gene expression analysis identified several defence-related genes that are highly expressed in the resistant genotypes relative to the susceptible genotype in response to *P. cactorum* after wounding. These included putative disease resistance (*R*) genes encoding receptor-like proteins, receptor-like kinases, nucleotide-binding sites, leucine-rich repeat proteins, RPW8-type disease resistance proteins, and ‘pathogenesis-related protein 1’. Seven of these *R*-genes were expressed only in the resistant genotypes and not in the susceptible genotype, and these appeared to be present only in the genomes of the resistant genotypes, as confirmed by PCR analysis. We previously reported a single major gene locus *RPc-1* (*Resistance to Phytophthora cactorum 1*) in *F. vesca* that contributed resistance to *P. cactorum*. Here, we report that 4–5% of the genes (35–38 of ca 800 genes) in the *RPc-1* locus are differentially expressed in the resistant genotypes compared to the susceptible genotype after inoculation with *P. cactorum*. In particular, we identified three defence-related genes encoding wall-associated receptor-like kinase 3, receptor-like protein 12, and non-specific lipid-transfer protein 1-like that were highly expressed in the resistant genotypes compared to the susceptible one. The present study reports several novel candidate disease resistance genes that warrant further investigation for their role in plant defence against *P. cactorum*.

## 1. Introduction

Crown rot, caused by the oomycete *Phytophthora cactorum* (Lebert & Cohn) J. Schröt, is a destructive soil-borne disease of strawberry (*Fragaria* spp.) [1]. The disease occurs sporadically, but serious crop losses are evident from previous outbreaks [2,3]. Although *P. cactorum* has more than 200 hosts among a wide range of plant species, crown rot of strawberry is caused by a distinct pathotype [4,5,6,7]. The pathogen forms sporangia that release motile zoospores, which infect the root system and the vascular bundles of the plant, resulting in symptoms such as brown necrosis, wilting, and death (in severe cases [8,9]). The pathogen may also infect fruits, which is a serious concern in strawberry jam production [10]. Although fungicides such as methyl bromide, mefenoxam, fluopicolide, and metalaxyl have been shown to be effective against *P. cactorum* [11,12,13,14], the phasing out of methyl bromide due to its effect on ozone depletion, the development of fungicide resistance, and strict regulations on the use of metalaxyl due to its potential risks to human and animal health present challenges in the management of both of these diseases in strawberry [14,15,16,17]. Thus, alternative approaches are needed to control these diseases in strawberry in a sustainable and ecofriendly way.

*Fragaria* species have a broad genetic background, ranging from diploid (2n = 2x = 14) to decaploid (2n = 10x = 70) [18]. Although most of the commercially grown octoploid (2n = 8x = 56) strawberry (*F.* × *ananassa* Duch.) cultivars are susceptible to crown rot, their wild diploid relatives carry useful resistance to *P. cactorum* [19,20,21]. The diploid *F. vesca* is considered a model plant for the Rosaceae family, mainly because of its short generation time, the availability of an efficient and facile in vitro regeneration and transformation system [22,23], and the available genome sequences [24,25,26]. Furthermore, the diploid level of *F. vesca* alleviates all genetic and molecular analyses relative to its octoploid descendant.

Plants defend themselves from invading pathogens through a multilayer immune system [27]. The pattern-triggered immunity includes several plant immune receptors, which are broadly classified as receptor-like kinases (RLKs) and receptor-like proteins (RLPs) [28]. These are plasma membrane-localised proteins that consist of a ligand-binding ectodomain that perceive the conserved structures of pathogen-associated molecular patterns (PAMPs). The perception of specific patterns/epitopes of microbes induces both local and systemic immune responses that contribute to broad-spectrum disease resistance in plants [29]. In an effector-triggered immunity, several cytoplasmic immune receptors recognise the secreted pathogen effectors and mediate signalling that involves disease resistance (R) proteins, which belong to the class of nucleotide-binding leucine-rich repeat (NLR) proteins. The NLR proteins have a conserved C-terminal leucine-rich repeat (LRR) domain fused to a central nucleotide binding site (NB-ARC) domain and a variable N-terminal domain [30]. NLR proteins with an N-terminal Toll/interleukin-1 receptor (TIR) domain are called TNLs (TIR-NB-ARC-LRR), and those with an N-terminal coiled-coil (CC) domain are called CNLs (CC-NB-ARC-LRR). More recently, RPW8-type CC-NLRs have been described—many were initially classified as CNLs but were refined based on their similarity to the CC disease resistance gene *RPW8* (*Resistance to Powdery Mildew 8*) [31,32]. Effector-triggered immunity is activated by either the direct or indirect binding of effectors by host NLRs [32,33]. Both pattern-triggered immunity and effector-triggered immunity result in localised cell death or a hypersensitive response (HR), which restrict the growth of biotrophic and hemi-biotrophic pathogens [34,35,36].

The strawberry defence system against pathogens, including *P. cactorum*, has been extensively reviewed and is similar to the defence systems of other plants [37,38]. In the last two decades, several studies have identified the genetic resources of resistance to *P. cactorum* in different strawberry accessions and genotypes [19,39,40,41,42]. Previous studies have identified several quantitative trait loci (QTLs) associated with resistance to *P. cactorum* in both diploid [43] and octoploid *Fragaria* spp. [44,45]. In addition, several resistance gene analogues (RGAs) of *F. vesca* were identified using a PCR-based approach called NBS profiling, and it showed an increased expression of RGAs during the early infection stage of *P. cactorum* [46]. Furthermore, Taljamo et al. [47] studied the transcriptome of a resistant *F. vesca* genotype, Hawaii4, in response to *P. cactorum* 48 h after inoculation and reported several defence-related genes upregulated in the *RPc-1* (*Resistance* to *Phytophthora cactorum 1*) locus (which was identified by Davik et al. [43]). We hypothesised that the use of both resistant and susceptible genotypes in transcriptomic studies is crucial for identification of the genes controlling resistance to *P. cactorum* in strawberry, as *R*-genes may be constitutively expressed. In the present study, we explored the transcriptomes of the following three *F. vesca* accessions (which we previously tested for their resistance to *P. cactorum* [19]): NCGR1603 and Bukammen, which are resistant; and NCGR1218, which is susceptible. Here, we report several novel candidate *R*-genes encoding RLPs, RLKs, NLRs, and RPW8-related disease resistance proteins in *F. vesca* against *P. cactorum* that was not reported earlier in the strawberry pathosystem.

## 2. Results

### 2.1. Transcriptome of Resistant and Susceptible Strawberry Genotypes Inoculated with P. cactorum

To investigate the molecular mechanism of defence against *P. cactorum* in *F. vesca*, the transcriptome of the susceptible genotype, NCGR1218, and two resistant genotypes, NCGR1603 and Bukammen, were compared 48 h after wounding and inoculation with *P. cactorum* (I48), and with their respective controls (wounded and inoculated with water-W48; untreated-C00). At the time of harvest (48 h), no visible symptoms were observed, but additional plants (inoculated and control plants), were kept for four weeks to study the disease progression. After two weeks, wilting and clear necrotic lesions were observed in the inoculated susceptible genotype NCGR1218 (Figure 1a). In contrast, the genotypes NCGR1603 and Bukammen had no visual symptoms of the disease four weeks after inoculation, except for a few patches of necrosis in the rhizomes in a small number of the inoculated plants (Figure 1a,b).

A total of 2,127,111,692 high-quality trimmed reads were de novo assembled to 412,970 transcript isoforms. The longest transcript isoforms of each gene were extracted, and subsequently annotated using the NCBI non-redundant (Nr) database. Using the *F. vesca* Hawaii v4.a1 database, 16,427 transcripts matched genes in the *F. vesca* Hawaii4 genome (Appendix A). These transcripts were assigned an individual *F. vesca* Hawaii v4.a1 gene ID and examined for differential gene expression, while the remaining unaligned transcripts were discarded from the analysis. Thus, each gene is represented by a specific transcript in the differential gene expression analysis.

### 2.2. Analysis of Differentially Expressed Genes in F. vesca Genotypes

The normalised counts of transcripts per million (TPM) were used to determine which genes were differentially expressed two-fold or more in the three *F. vesca* genotypes. Only genes with an average TPM ≥ 1 in at least one of the samples were included. As a result, 1645 and 1768 differentially expressed genes (DEGs) were detected in the inoculated resistant genotypes, NCGR1603 and Bukammen, respectively, when compared to the inoculated susceptible genotype NCGR1218 48 h after inoculation with *P. cactorum* (I48) (Appendix A). The comparison of inoculated genotypes (I48) with their wounded controls (W48) revealed 1391, 322, and 193 DEGs in NCGR1218, NCGR1603, and Bukammen, respectively. Similarly, 1660, 979, and 862 DEGs were detected in the inoculated NCGR1218, NCGR1603, and Bukammen (I48), respectively, when compared to their untreated controls (C00) (Appendix A).

The level of variance in gene expression patterns between the genotypes and replicates was analysed using a multidimensional scaling (MDS) plot (Figure 2). The expression patterns of each genotype not only changed in response to inoculation with *P. cactorum* after wounding (I48), but also after wounding alone. The two resistant genotypes NCGR1603 and Bukammen were clearly separated from the susceptible genotype NCGR1218. In addition, both the resistant genotypes were clustered relatively closely, indicating similarity in gene expression after inoculation with *P. cactorum* (Figure 2). The four biological replicates from each genotype and treatment were clustered closely, and the inoculated samples were well separated from the controls. 

### 2.3. GO Term Enrichment Analysis of DEGs

To understand the broad functions of the DEGs identified in the inoculated resistant genotypes compared to the susceptible genotype, WEGO was used to categorise and analyse all DEGs into three main categories: the cellular component (CC), molecular function (MF), and biological process (BP) of the GO classification. A total of 1082 DEGs (483 upregulated and 599 downregulated genes) out of 1645 DEGs and a total of 1105 DEGs (479 upregulated and 626 downregulated) out of 1768 DEGs had at least one GO term for each gene in the inoculated resistant genotypes NCGR1603 and Bukammen, respectively, relative to the inoculated susceptible genotype NCGR1218. The WEGO analysis categorised the 1082 DEGs of NCGR1603 and the 1105 DEGs of Bukammen into 52 and 58 functional categories, respectively (Appendix A). The top three functional categories with the highest percentage of genes were involved in catalytic activity (CC category), metabolic process (BP category), and binding (MF category) in both of the resistant genotypes (Appendix A). In both the genotypes, almost equal proportions of upregulated and downregulated genes were associated with the different functional classes.

To identify GO terms that were significantly different in number between upregulated and downregulated genes, a GO enrichment analysis was performed with a *p*-value threshold of less than 0.05. The 1082 DEGs in NCGR1603 were significantly enriched in six main functional categories, whereas the 1105 DEGs in Bukammen were significantly enriched in thirteen main functional categories (Figure 3). Five functional categories were enriched in both the resistant genotypes: catalytic activity, acting on DNA (GO:0140097); catalytic activity, acting on RNA (GO:0140098); response to stress (GO:0006950); macromolecule localisation (GO:0033036); and localisation (GO:0051179) (Figure 3).

### 2.4. Genes Involved in Defence against P. cactorum

Genes upregulated 48 h after inoculation with *P. cactorum* were examined for their potential role in the defence against the pathogen by searching for known resistance proteins in the annotations. Twenty-three defence-related genes were expressed only in the two resistant genotypes and not in the susceptible one (Table 1, Appendix A). These genes encode proteins that have a putative role in plant defence against biotic stresses—including fungi, oomycetes, viruses, bacteria, and insects—or against abiotic stresses (Table 1).

Thirty-one genes encoding putative disease resistance proteins were differentially expressed in the resistant *F. vesca* genotypes compared to the susceptible genotype (Figure 4). These included genes encoding the following: receptor-like proteins (RLPs); receptor-like kinases (RLKs); proteins with nucleotide-binding sites and leucine-rich repeats (NLRs), grouped as CNLs (CC-NB-ARC-LRR); TNLs (TIR-NB-ARC-LRR); and RPW8-related disease resistance proteins. Interestingly, seven of these *R*-genes were not expressed in the susceptible genotype at all (Figure 4; Table 1; Appendix A). Moreover, 5 of the 31 putative *R*-genes (FvH4_6g49500.1, FvH4_1g15240.1, FvH4_7g31270.1, FvH4_6g11080.1, FvH4_6g32690.1) were upregulated in the resistant genotypes after wounding alone relative to the untreated control, while two genes putatively encoding an NLR protein (FvH4_6g18970.1) and an RLK (FvH4_4g22030.1) were upregulated in response to inoculation with *P. cactorum.* The remaining putative *R*-genes were constitutively expressed in the resistant genotypes, irrespective of wounding or pathogen inoculation (Appendix A).

### 2.5. Expression of Defence-Related Genes in the RPc-1 Locus

The *RPc-1* locus, which we previously reported as being involved in the resistance to *P. cactorum* in strawberry [43], was examined, in particular, to identify differentially expressed genes in this locus. The 3.3 Mb *RPc-1* locus contains approximately 800 genes, including 69 with a potential role in disease resistance [43]. Several genes detected in this locus were upregulated in the resistant genotypes after inoculation with *P. cactorum* (Appendix A). Among the 11–12 upregulated genes in the resistant genotypes, 9–11 were shared by the susceptible genotype NCGR1218 when compared to their untreated controls (Figure 5a; Appendix A). Comparisons between the susceptible genotype NCGR1218 and the resistant genotypes NCGR1603 and Bukammen revealed 35 and 38 DEGs, respectively, which represent 4–5% of the total genes in the *RPc-1* locus (Appendix A).

Twenty-seven DEGs were shared by the resistant genotypes, relative to the susceptible genotype (Figure 5b). Based on the functional annotations of the 27 DEGs in *RPc-1*, three of these were defence-related genes. These included genes encoding wall-associated receptor-like kinase 3 (WAK3-like, FvH4_6g15920.1), receptor-like protein 12 (RLP12, FvH4_6g11080.1), and non-specific lipid-transfer protein 1-like (nsLTP1-like, FvH4_6g09980.1). The expression of the *WAK3-like* gene was 39- and 33-fold higher in the inoculated resistant genotypes- NCGR1603 and Bukammen, respectively, than in the inoculated susceptible genotype NCGR1218 (Appendix A). The gene exhibited a constitutive expression pattern in the resistant genotypes, irrespective of their inoculation with *P. cactorum*. The *RLP12* gene showed 2.9- and 2.7-fold higher expression in the inoculated NCGR1603 and Bukammen, respectively, than in NCGR1218, while *nsLTP1-like* gene had 2.3-fold higher expression in both the resistant genotypes compared with the inoculated NCGR1218 (Appendix A).

### 2.6. Differential Expression Analysis of Transcription Factor Genes

The differential expression of transcription factor genes was examined in the three *F. vesca* genotypes after inoculation with *P. cactorum*. A total of 104 and 95 DEGs encoding putative transcription factors (TFs) were detected in the susceptible genotype NCGR1218 after inoculation with *P. cactorum*, compared to the controls C00 and W48h, respectively (Appendix A).

Transcriptional changes were also detected for nine TF families in the resistant genotypes after wounding, but few changes were observed in response to *P. cactorum* (Appendix A). The majority of these genes belonged to the following TF families: AP2/ERF (APETALA2/ethylene-responsive element-binding factor), WRKY (conserved WRKYGQK amino acid sequence), and MYB (myeloblastoma) (Appendix A).

A comparison between inoculated susceptible and resistant genotypes revealed that most (37–39%) of the highly expressed TF genes in the resistant genotypes belonged to the C2H2 (CYS2-HIS2) zinc-finger domain-containing TF family (Figure 6; Appendix A). In contrast, genes encoding ethylene-responsive transcription factors (with the AP2/ERF domain), WRKY transcription factors, and MYB-related transcription factors were less expressed in the inoculated resistant genotypes than in the inoculated susceptible genotype (Figure 6; Appendix A).

### 2.7. Flavonoid, Isoprenoid and Phytohormone Signalling Pathway Genes

Genes encoding enzymes in the flavonoid and isoprenoid biosynthetic pathways that are often involved in defence were differentially expressed in all three *F. vesca* genotypes after inoculation with *P. cactorum* (Appendix A). The number of DEGs belonging to these two pathways were higher (11–18 DEGs) in the susceptible genotype than in the resistant genotypes (4–11 DEGs). A higher average fold change in gene expression was observed for the upregulated DEGs compared with the downregulated DEGs in all three *F. vesca* genotypes (Appendix A). Genes connected to the phytohormone signalling pathways ethylene (EA), jasmonic acid (JA), and salicylic acid (SA) were upregulated in the inoculated susceptible genotype relative to its controls (C00 and W48) (Appendix A). In contrast, only a few of these were differentially expressed in the resistant genotypes. Around 60% of the genes connected to the auxin-related pathway were downregulated in the susceptible genotype in response to inoculation with *P. cactorum* after wounding (Appendix A).

### 2.8. Pathogenesis-Related Genes

Many genes encoding putative pathogenesis-related (PR) proteins were upregulated in all of the *F. vesca* genotypes in response to inoculation with *P. cactorum* after wounding. These included genes encoding pathogenesis-related protein 1 and a basic form of pathogenesis-related protein 1 (PR1-family), glucan endo-1,3-beta glucosidases and endo-1,3-beta-D-glucanases (PR2-family), chitinases and endo-chitinases (PR3-family), proteins with BARWIN-like domains (PR4-family), proteinase inhibitors (PR6-family), peroxidases (PR9-family), and members of the PR10-family with unknown function (Appendix A). Interestingly, the expression of a gene encoding a basic form of a PR1 protein (FvH4_2g02880.1) was 5.8- 6.5-fold higher in the inoculated resistant genotypes NCGR1603 and Bukammen than in the inoculated susceptible genotype NCGR1218 (Appendix A). In the latter, the gene was downregulated in response to the inoculation (Appendix A).

### 2.9. Validation of Genes Expressed Only in the Resistant Genotypes

A selected number of genes that were expressed only in the resistant genotypes (Appendix A) were validated by semi-quantitative RT-PCR. These included genes putatively encoding a transmembrane protein (FvH4_5g24630.1); a galactose oxidase with kelch/beta-propeller domain (FvH4_1g22440.1); a zinc finger BED domain-containing protein, DAYSLEEPER (FvH4_7g20440.1); three NLR genes (FvH4_5g16110.1, FvH4_5g16070.1 and FvH4_6g18970.1); a transcription factor C2H2 family (FvH4_6g34080.1); and an uncharacterised protein (FvH4_1g22450.1). The full-length transcripts of these genes were amplified (Figure 7), and their expression patterns confirmed the RNA-Seq data (Appendix A).

### 2.10. Detection of Genes Distinct for the Resistant Genotypes

Twelve genes that were expressed only in both of the resistant genotypes were randomly selected and tested for their presence or absence in the genomes of three *F. vesca* genotypes, NCGR1218, NCGR1603, and Bukammen, and a moderately susceptible strawberry (*F.* × *ananassa*) cultivar, Korona, by PCR amplification (Appendix A). The moderately susceptible Korona cultivar was included to compare the presence/absence polymorphisms of the susceptible genotype NCGR1218. Seven of these genes were detected only in the genomes of the resistant genotypes NCGR1603 and Bukammen, and not in NCGR1218 and Korona (Figure 8).

These included genes putatively encode a predicted membrane localized protein with unknown function (FvH4_5g24630.1); a galactose oxidase with kelch/beta-propeller domain-containing protein (FvH4_1g22440.1); a zinc finger BED domain-containing protein, DAYSLEEPER (FvH4_7g20440.1); a transcription factor C2H2 family (FvH4_6g34080.1); an uncharacterised protein (FvH4_1g22450.1); and disease resistance RPP13-like protein 1 (FvH4_5g16110.1) (Figure 8a,b). Another gene encoding disease resistance RPP13-like protein 1 (FvH4_5g16070.1) was also detected in the resistance genotypes, but, for this gene, the susceptible genotype NCGR1218 showed a faint band slightly below the expected amplicon size for one of the primer pairs tested, which likely resulted from the amplification of a pseudogene (Figure 8a).

## 3. Discussion

In this study, massive transcriptional changes were observed in *F. vesca* genotypes in response to inoculation with *P. cactorum* after wounding. The number of DEGs in the inoculated susceptible genotype were relatively higher than in the inoculated resistant genotypes when compared to their untreated and wounded controls. The treatment with wounding alone resulted in intense transcriptional reprogramming in the resistant genotypes compared to their untreated controls. In contrast, little variation in gene expression was observed in the susceptible genotype after wounding, which is illustrated by the close clustering of the biological replicates of the wounded treatment and the untreated control (Figure 2). This is not unexpected, as previous studies have also shown that different plant species share common sets of genes in response to wounding and pathogen attack [74,75,76]. Similar transcriptional responses were also observed in a comparison of susceptible and resistant wild tomato (*Solanum pimpinellifolium*) accessions in response to *P. parasitica* [77].

GO analysis indicated that the number of DEGs in both the resistant genotypes was higher in the three functional categories, namely, catalytic activity (CC category), metabolic process (BP category), and binding (MF category). These results are consistent with other studies on plant pathogens in, e.g., eucalyptus [78], tobacco [79], and wheat [80].

Several putative disease resistance genes that have a potential role in plant defence in *F. vesca* were identified. These include genes encoding RLPs, RLKs, NLRs (including TNLs and CNLs), and RPW8-type disease resistance proteins. Most of these genes were highly expressed in the two resistant genotypes relative to the susceptible genotype 48 h after inoculation with *P. cactorum.*

Some of these putative *R*-genes were only expressed in the resistant genotypes (Figure 4; Appendix A). Interestingly, several defence-related genes were detected only in the genomes of the resistant genotypes (Figure 8), explaining the lack of expression in the susceptible genotypes. The presence of these defence-related genes in the resistant genotypes indicate that they are important for crown rot resistance in strawberry. However, most of the putative *R*-genes (26 out of a total of 31 highly expressed genes in the resistant genotypes) were constitutively expressed, irrespective of pathogen inoculation. These results suggest that the defence mechanisms of the resistant genotypes of *F. vesca* are activated prior to pathogen infection, as observed earlier against *Botrytis cinerea* in a resistant genotype [81]. Two of the differentially expressed *R*-genes (FvH4_1g15330.1 and FvH4_2g36850.1) were identified earlier (RGA2 and RGA194, respectively) and followed a similar pattern of expression as reported previously [46]. The newly identified *R*-genes FvH4_6g18970.1 (NLR) and FvH4_4g22030.1 (RLK) are interesting as they were upregulated in the resistant genotypes after wounding and inoculation with *P. cactorum* (Appendix A). Further functional analysis is required to pinpoint the individual or collective role of these *R*-genes in conferring resistance to *P. cactorum*.

Besides the classical *R*-genes, several defence-related genes were only expressed in the inoculated resistant genotypes (Table 1). For example, genes putatively encoding a DAYSLEEPER protein with a zinc finger BED domain, and a *Clostridium* epsilon toxin ETX/MTX2 with an agglutinin domain, could play a role in the defence against *P. cactorum*. Such proteins have previously been found to confer resistance to fungal pathogens [55,57]. For example, Marchal et al. [55] demonstrated that three NLRs containing a noncanonical zinc finger BED domain in the N-terminal confer broad spectrum resistance against the stripe rust fungus *Puccinia striiformis* f. sp. *tritici* in wheat. Moreover, BED domains from non-NLR DAYSLEEPER proteins possess DNA binding activity in *Arabidopsis* [56] and, therefore, may influence global gene expression to enhance resistance.

In a previous study of a diallel cross between the resistant *F. vesca* genotype Bukammen and the susceptible Haugastøl 3, we identified a single major gene locus, *RPc-1*, that was attributed to *P. cactorum* resistance [43]. In the present study, genes in this locus were differentially expressed in all three studied genotypes 48 h after inoculation with *P. cactorum*, compared to their respective controls (C00 and W48) (Figure 5, Appendix A). Some genes in *RPc-1* were also differentially expressed in both the resistant genotypes NCGR1603 and Bukammen, relative to the susceptible genotype NCGR1218 after inoculation (Figure 5b). Ten DEGs detected in the *RPc-1* locus were previously reported to be upregulated in the transcriptome study of the Hawaii4 genotype which is quite resistant to *P. cactorum* [47]. However, 6 out of these 10 upregulated defence-related genes in the Hawaii4 transcriptome—FvH4_6g09300.1, FvH4_6g13200.1, FvH4_6g10510.1, FvH4_6g15760.1, FvH4_6g11660.1, and FvH4_6g09640.1 (NCBI accessions: LOC101295534, LOC101310048, LOC101311683, LOC101309855, LOC101290881, and LOC101312550) [47]—were also upregulated in the susceptible genotype NCGR1218 in our study, suggesting that these genes may participate in defence but do not fully control resistance to *P. cactorum.* Our data on the *RPc-1* locus genes signify that the use of single resistant genotypes in RNA-Seq without a susceptible host for comparison can lead to false positive results or can potentially mask candidate disease resistance genes.

Interestingly, the *ns-LTP1-like* gene (FvH4_6g09980.1/LOC101301595) from the *RPc-1* locus that was upregulated in the Hawaii4 transcriptome [33] was upregulated by more than two-fold in the resistant genotypes compared to the susceptible genotype after inoculation with *P. cactorum*. The present study also identified an additional *RLP12* gene (FvH4_6g11080.1) in *RPc-1* whose expression levels were significantly higher (*p* < 0.05) in the resistant genotypes than in the susceptible one. Previous studies have shown that ns-LTPs and RLPs play a crucial role in plant defence against several biotic stresses, including bacteria, fungi, oomycete, viruses, and insects [82,83,84,85,86,87,88]. The exogenous application of antimicrobial nsLTPs was even shown to inhibit pathogen growth in vitro [86]. In addition to the *nsLTP1-like* and *RLP12* genes, a *WAK* gene (FvH4_6g15920.1) showed a more than 30-fold higher level of expression in both the resistant genotypes than in the susceptible genotype (Appendix A). Furthermore, the *WAK* gene was constitutively expressed in both the control and the inoculated resistant genotypes, suggesting its potential role in strawberry basal immunity against several biotrophs, hemi-biotrophs, and necrotrophic pathogens, as reported for such genes in other plant species [89,90,91]. The three abovementioned *RPc-1* genes in the resistant *F. vesca* genotypes are promising candidates that require further investigation to uncover their role in the defence against *P. cactorum*. The present findings of differentially expressed defence-related genes in the *RPc-1* locus support its involvement in the defence against *P. cactorum*.

In the present study, several TF genes were differentially regulated during the strawberry–*P. cactorum* interaction (Figure 6, Appendix A). Highly upregulated TF genes in the inoculated susceptible genotype relative to its controls belonged to the ethylene-responsive transcription factor (AP2/ERF) and WRKY transcription factor families (Appendix A). A similar expression pattern of these TF gene families was previously observed after infection by *P. parasitica* in *Nicotiana benthamiana* and *A. thaliana*, both of which are susceptible host plants [92,93]. These TF gene families were also upregulated in the inoculated resistant genotypes relative to their untreated control. However, only minor changes were observed in the inoculated samples relative to their wounded mock control for these genotypes. The number of expressed genes encoding C2H2 Zinc-finger RING-type TFs was much higher in the inoculated resistant genotypes than in the inoculated susceptible genotype (Figure 6, Appendix A), and most of these TF family genes were constitutively expressed in the resistant plants. This is compatible with previous studies pointing to C2H2-type zinc-finger TFs as positive regulators of plant defence against biotic and abiotic stresses (reviewed by Kiełbowicz-Matuk 2012) [94]. For example, overexpression of a zinc-finger transcription factor gene (*CAZFP1*) from pepper in *Arabidopsis* enhanced resistance to *Pseudomonas syringae* pv. *tomato* and simultaneously improved drought tolerance [95]. Similarly, overexpression of a Q-type zinc finger transcription factor gene (*StZFP2*) in potato conferred resistance to *P. infestans* [96]. The genes encoding C2H2 zinc-finger RING-type TFs of *F. vesca* require further attention as they are possibly involved in the defence against *P. cactorum*.

Several *PR* genes were also differentially expressed in the three *F. vesca* genotypes in response to inoculation with *P. cactorum*. Interestingly, a gene (FvH4_2g02880.1) encoding a basic form of PR1 protein was upregulated in the resistant genotypes but downregulated in the susceptible genotype after wounding and inoculation with *P. cactorum* (Appendix A). Expression of basic *PR1* genes have previously been observed in response to *Magnaporthe grisea* in rice and *Tobacco mosaic virus* in tobacco [97,98]. Previous studies have reported that PR-1 proteins exhibit antimicrobial activity against *Phytophthora* species [99,100], indicating that the role of the *F. vesca PR1* gene (FvH4_2g02880.1) in the defence against *P. cactorum* is worth investigating.

Flavonoid and isoprenoid compounds have been documented as defence signalling molecules against several phytopathogens [37]. In this study, DEGs connected to flavonoid and isoprenoid pathways were numerically higher in the inoculated susceptible genotype than in the resistant genotypes (Appendix A). However, most of these genes were upregulated in all three *F. vesca* genotypes in response to *P. cactorum*, which is consistent with a previous transcriptome study on the Hawaii4 genotype [47]. A similar expression pattern was also observed for the genes involved in the phytohormone signalling pathways. EA, JA, and SA hormonal signalling pathways have also been reported in plant defence against other *Phytophthora* species [101,102]. Genes connected to JA and SA signalling pathways were upregulated in the susceptible genotype in response to *P. cactorum*; however, small changes in gene expression were observed in the resistant genotypes (Appendix A). This is possibly due to the expression of *R*-genes in the resistant genotypes that may have restricted the growth of *P. cactorum* in the primary infected cells and, therefore, minimised SA and JA signalling responses to neighbouring cells. Most of the genes connected to auxin biosynthesis and transport were more downregulated in the susceptible genotype relative to the untreated and wounded controls, compared with the resistant genotypes. However, some of these genes were also downregulated in the inoculated resistant genotypes. In the strawberry–*P. cactorum* pathosystem, the downregulation of auxin-related genes could act as an indicator of disease susceptibility in the genotype NCGR1218. It is known that the impact of auxin on plant defence can be positive or negative depending on the pathogen biology and crosstalk with other hormonal pathways [103]. In two independent *Arabidopsis* studies, suppression of the auxin response pathway enhanced susceptibility to the hemibiotrophic *Phytophthora cinnamomi* and two necrotrophic fungi [104,105]. Conversely, overexpression of auxin biosynthesis gene *YUCCA1* in Arabidopsis exhibited increased susceptibility to the bacterial pathogen *Pseudomonas syringae* [106]. Thus, future functional analysis is required to understand the role of auxin in mediating defence signalling in strawberry against *P. cactorum*.

## 4. Materials and Methods

### 4.1. Plant Material and Pathogen Inoculation

Clonally propagated plants were grown in a greenhouse at 18 °C and under a 16/8 h light/dark regime. For pathogen inoculation, *P. cactorum* isolate 10300 [107] was grown on vegetable juice (V8) agar plates and incubated in the dark for two weeks at room temperature (~20 °C). Zoospore suspensions were prepared as described by Eikemo et al. [108]. Plants were gently wounded in the rhizome (crown) with a sterile scalpel and inoculated with 2 mL of zoospore suspension (2 × 10^5^ spores/mL) or water (mock control). Four biological replicates were used for each of the treatments (inoculated-I48 and mock-treated-W48) and for the untreated controls (C00), where each replicate consisted of four individual plants from each genotype. Samples were harvested from the rhizome at the time of inoculation and after 48 h. Sampling was performed at 48 h after inoculation as this represents the early infection stage of *P. cactorum* based on a previous temporal expression study of defence-related genes in resistant and susceptible *F. vesca* genotypes and findings of visible hyphae on the *F. vesca* root surfaces 48 h after inoculation with *P. cactorum* [46,47]. The samples were flash frozen in liquid nitrogen and stored at −80 °C until RNA isolation. Additional plants, inoculated and control plants, were kept for four weeks after inoculation to study the disease progression.

### 4.2. Disease Scoring and Statistical Analysis

Plants were scored for disease symptoms on a scale from 1 to 8, as described previously by Bell et al. [48]. One-way ANOVA with post hoc Tukey’s HSD (honestly significant difference) test was performed to calculate the significant level of resistance among the strawberry genotypes.

### 4.3. RNA Isolation, Library Preparation and Sequencing

Total RNA was isolated from strawberry rhizomes using the Spectrum^TM^ Plant Total RNA Kit (Sigma-Aldrich, St. Louis, MO, USA) according to the manufacturer’s instructions. In addition, 30 min on-column DNase digestion was performed on the isolated RNA to remove traces of DNA contamination (Sigma-Aldrich, USA). The quantity and quality of the isolated RNA was assessed using the Nanodrop 2000 Spectrophotometer (Thermo Scientific, Waltham, MA, USA), and the Agilent RNA 6000 Nano kit in the Agilent 2100 Bioanalyzer (Agilent Technology, Santa Clara, CA, USA), respectively. The RNA integrity numbers (RIN) for the isolated RNA samples ranged from 6.8 to 9.5 among the 36 samples that were used for library preparation and sequencing. The libraries were prepared using the TruSeq^TM^ stranded total RNA library prep kit (Illumina), and the samples were indexed and sequenced using four lanes on an Illumina HiSeq 3/4000 (2 × 150 bp) System by the Norwegian Sequencing Centre, Oslo, Norway.

### 4.4. De Novo Transcriptome Assembly and Data Analysis

The transcripts were de novo assembled instead of being mapped to a reference genome (*F. vesca* v4.0.a1 Hawaii 4) to avoid the potential loss of novel or unique transcripts, genotype-specific sequences, and to recover transcripts of *P. cactorum* 10300.

Following the RNA-Seq, the resulting sequence files (four forward, four reverse) from each sample were concatenated to one forward and one reverse fastq file. Raw reads were trimmed for adaptor sequences, ambiguous bases and quality filtered using Trimmomatic v0.38 (phred33, ILLUMINACLIP:TruSeq3-PE.fa:2:30:10 LEADING:3 TRAILING:3 SLIDINGWINDOW:4:15) [109]. The trimmed reads were assembled using Trinity v2.8.4, and transcript quantification of the different samples was performed using the pseudo-alignment method Kallisto [110]. The transcripts were normalised using the CLC Genomics Workbench v11.01 (QIAGEN, Aarhus, Denmark) with the TPM (transcripts per million) normalisation method [111]. Normalised transcript counts that were greater than or equal to 1 TPM in at least one of the samples under comparison were analysed for differential expression. Differential expression (DE) analysis was performed in the CLC Genomics Workbench. Transcripts were assigned a *Fragaria vesca* v4.a.1 gene ID using reciprocal BLAST (blastx and tblastn) with an expectation value (E) < 10^−10^ as a threshold. A *p*-value < 0.05 was used as a selection criterion for differentially expressed genes (DEGs) between the genotypes and with their controls. A fold change (FC) criterion ≥2 or ≤−2 was used to define upregulated and downregulated genes. Transcripts were annotated using Blast2Go v5.0 [112]. GO enrichment analyses of DEGs were performed using WEGO 2.0 [113]. KEGG orthologs and InterPro domain annotations were adapted from the Rosaceae database (www.rosaceae.org (accessed on 25 June 2023)). Comparisons of the differentially expressed genes in the *RPc-1* locus were performed using VennPlex v1.0.0.2 [114].

### 4.5. RNA-Seq Data Validation by Semi-Quantitative Reverse Transcription-PCR (RT-PCR)

Semi-quantitative RT-PCR analysis was used to validate RNA-Seq data for genes that had zero expression level in the susceptible genotype and a detectable transcript level in the resistant genotypes. Briefly, the complementary DNA (cDNA) was synthesised from 1 µg of total RNA isolated from each of the samples, using the Transcriptor High Fidelity cDNA Synthesis Kit (Roche, Germany). Following reverse transcription, PCR was performed using 1.5 µL of template cDNA in a reaction mix that contained 5 µL of 5X Phusion^®^ HF Buffer (New England BioLabs, Ipswich, MA, USA), 1.6 µL of 2.5 mM dNTPs, 1.25 µL of 10 mM of each of the forward and reverse primers, 0.75 µL of 100% DMSO, and 0.25 µL of Phusion^®^ High-Fidelity DNA Polymerase (New England BioLabs, Ipswich, USA) in a total volume of 25 µL (adjusted using deionised water). The PCR conditions were 95 °C for 5 min, followed by 35 cycles of 95 °C for 1 min, 60 °C for 30 s, 72 °C with a varying elongation time depending on the amplicon size (30 s for <1 kb; 1 min for 1–2 kb; 2 min for 2–4 kb), and a final 7 min extension at 72 °C (Appendix A). The housekeeping gene *Elongation factor 1 alpha* (*EF1a*) was used as an internal control [115]. Five microliters of the PCR products were loaded onto a 1% agarose gel containing ethidium bromide. Gel electrophoresis was performed at 130 V for 40 min and gel images were recorded under UV light.

### 4.6. Isolation of Genomic DNA and PCR Analysis

Genomic DNA was isolated from the strawberry leaves of ten individual plants from each genotype, as described by Nunes et al. [116] with slight modifications. Briefly, 100 mg of the crushed sample was dissolved in 900 µL of CTAB buffer solution (2% cetyltrimethylammonium bromide, 100 mM of tris(hydroxymethyl)aminomethane hydrochloride pH 8.0, 25 mM of ethylenediaminetetraacetic acid, 1 M of sodium chloride, 2% polyvinylpolypyrrolidone; Sigma-Aldrich, USA), and 4 µL of RNase A (Qiagen, Hilden, Germany) were added during the initial incubation period. The DNA precipitation step using isopropanol lasted for 1 h and was performed at −20 °C.

PCR was performed using 500 ng of genomic DNA in a reaction that included 2 µL of 10X AmpliTaq Buffer (Applied Biosciences, Waltham, MA, USA), 1.6 µL of 2.5 mM dNTPs, 0.5 µL of 10 µM forward and reverse gene-specific primers (Appendix A), and 0.1 µL of 5 U/µL AmpliTaq polymerase (Applied Biosciences, USA) in a total volume of 20 µL (adjusted using deionised water). The PCR conditions were 95 °C for 5 min, followed by 38 cycles of 95 °C for 1 min, 55–60 °C for 30 s, and 72 °C for 1 min. The *EF1a* gene was used as a positive control [115]. To minimise possible misinterpretations resulting from potential mutations in the primer binding sites, two sets of PCR primer pairs were tested for each gene in the *F. vesca* genotypes and in the *F*. × *ananassa* cultivar Korona, which is moderately susceptible (intermediate) to *P. cactorum* [89].

## 5. Conclusions

Overall, the comparative transcriptome analysis identified several novel defence-related genes that warrant further investigation for their individual or collective role in the strawberry defence against *P. cactorum*. These include genes encoding an NLR protein (FvH4_6g18970.1), an RLK (FvH4_4g22030.1), basic pathogenesis-related protein 1 (FvH4_2g02880.1), and other constitutively expressed *R*-genes, including the three *RPc-1* locus genes (*WAK3-like*, FvH4_6g15920.1; *RLP12*, FvH4_6g11080.1; and *nsLTP1-like*, FvH4_6g09980.1). Future functional studies are required to validate the role of these genes in controlling resistance to *P. cactorum* in strawberry and other hosts.

The use of susceptible and resistant strawberry genotypes in the transcriptome study enabled fine resolution of candidate resistance genes that are often masked in transcriptome studies performed with a single genotype. The majority of the *R*-genes, transcription factor genes, and some *PR* genes showed a constitutive expression pattern in the resistant genotypes, signifying that resistance in wild strawberry is activated prior to pathogen infection. The present study provides a new data resource and a theoretical basis to explore the resistance repertoire of wild strawberry against *P. cactorum*.

## Figures and Tables

**Figure 1 ijms-24-10851-f001:**
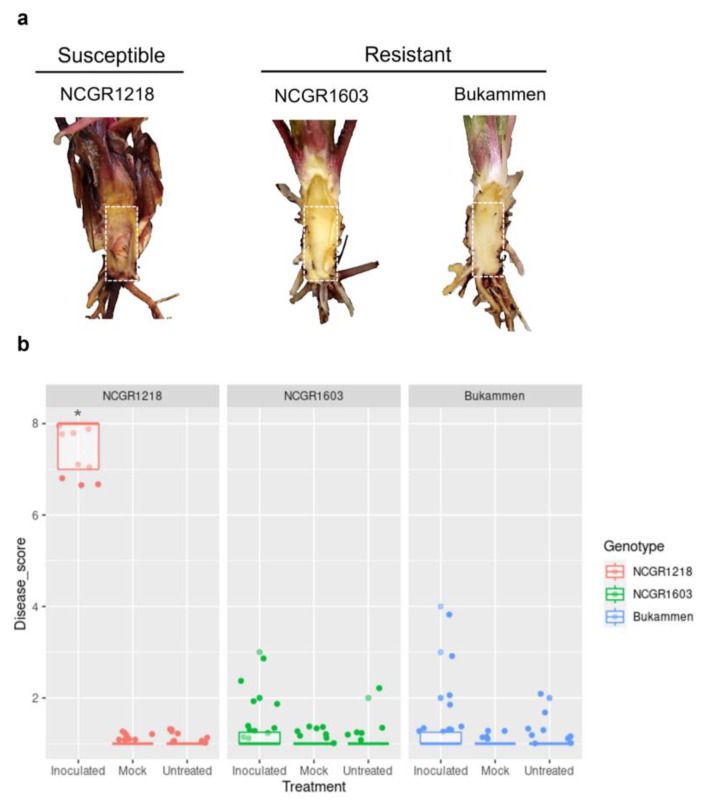
*Fragaria vesca* genotypes NCGR1218, NCGR1603, and Bukammen after inoculation with *Phytophthora cactorum*. (**a**). Phenotyping of crown rot resistance in the three *F. vesca* genotypes. The image of the susceptible genotype NCGR1218 was taken two weeks after inoculation with *P. cactorum*, while the images of the resistant genotypes NCGR1603 and Bukammen were taken four weeks after inoculation. Rectangular boxes with dashed white lines represent the strawberry rhizomes used for scoring disease development. (**b**). The disease scores of three *Fragaria vesca* genotypes NCGR1218, NCGR1603, and Bukammen after inoculation with *Phytophthora cactorum*. The scores are based on visual symptom observations, where scores 8, 7, 6, and 5 represent plants that showed wilting and collapsing of the whole plant during the first, second, third, and fourth weeks after inoculation, respectively, whereas scores <5 represent different degrees of necrosis visually observed in the rhizome for plants that survived four weeks after inoculation: 4 = clear necrosis covering at least 50% of the rhizome area; 3 = small patches of necrosis; 2 = minor brown/dark speckles; 1 = no symptoms as previously described [19,48]. Untreated and mock-inoculated (wounded and inoculated with water) plants were used as controls. The data represent the mean disease score of sixteen biological replicates for each genotype. The asterisk indicates a significant difference in the mean disease score between the genotype NCGR1218 and the two genotypes NCGR1603 and Bukammen after inoculation with *P. cactorum* (*p*-value < 0.01).

**Figure 2 ijms-24-10851-f002:**
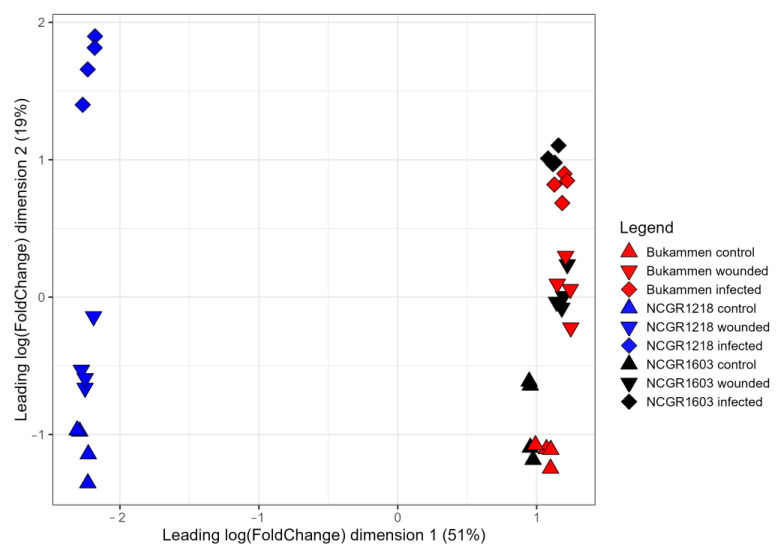
Multidimensional scaling plot of differentially expressed genes in the three *Fragaria vesca* genotypes NCGR1218, NCGR1603, and Bukammen in response to *Phytophthora cactorum*. The samples were harvested from rhizome tissue inoculated with *P. cactorum* after wounding and harvested 48 h later (infected); mock control (wounded); and untreated control (control). Each symbol in the plot represents a biological replicate.

**Figure 3 ijms-24-10851-f003:**
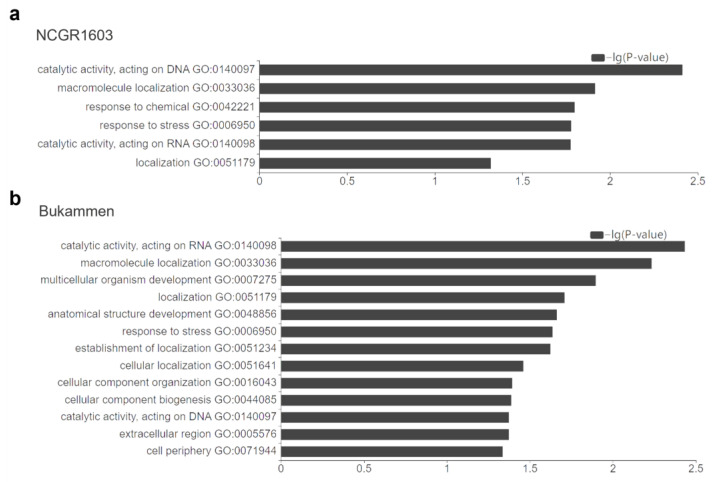
GO enrichment analysis of the differentially expressed genes (DEGs) in the resistant *F. vesca* genotypes (**a**) NCGR1603 and (**b**) Bukammen, relative to the inoculated susceptible genotype NCGR1218, 48 h after inoculation with *Phytophthora cactorum* (*p*-value < 0.05, Pearson chi-square test).

**Figure 4 ijms-24-10851-f004:**
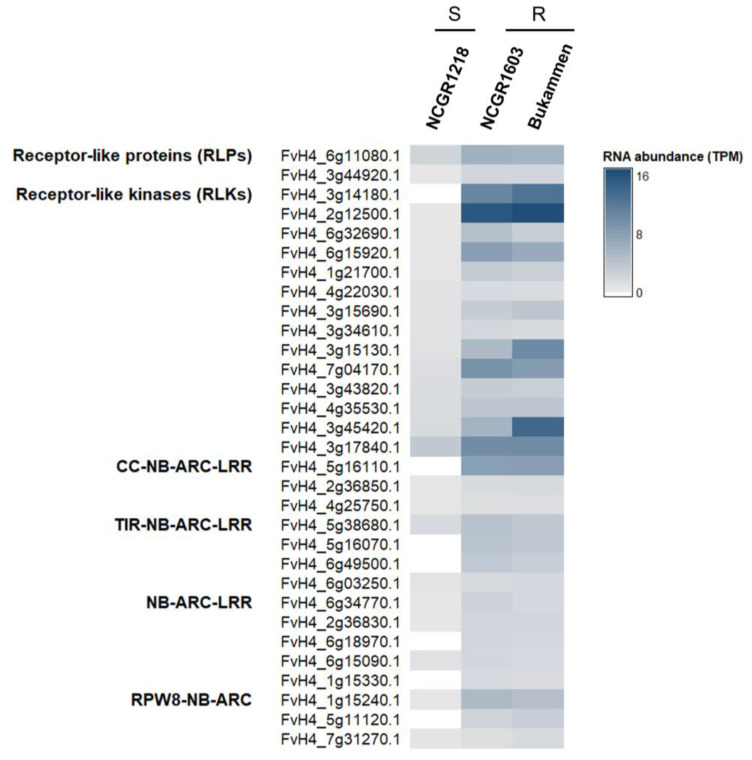
Expression levels of the 31 putative disease resistance genes in three *Fragaria vesca* genotypes 48 h after inoculation with *Phytophthora cactorum*. Genes that were uniquely expressed, or two-fold or more upregulated (*p* < 0.05), in the resistant genotypes (NCGR1603 and Bukammen) relative to the susceptible genotype (NCGR1218) are shown. The susceptible genotype NCGR1218 is indicated by S, and the two resistant genotypes NCGR1603 and Bukammen are indicated by R. A blue to grey colour gradient indicates transcript abundance in terms of transcripts per million (TPM), while white indicates no expression. The TPM values are the mean of the four biological replicates, each consisting of four individual plants for each genotype and treatment (in total, 16 plants per genotype and treatment).

**Figure 5 ijms-24-10851-f005:**
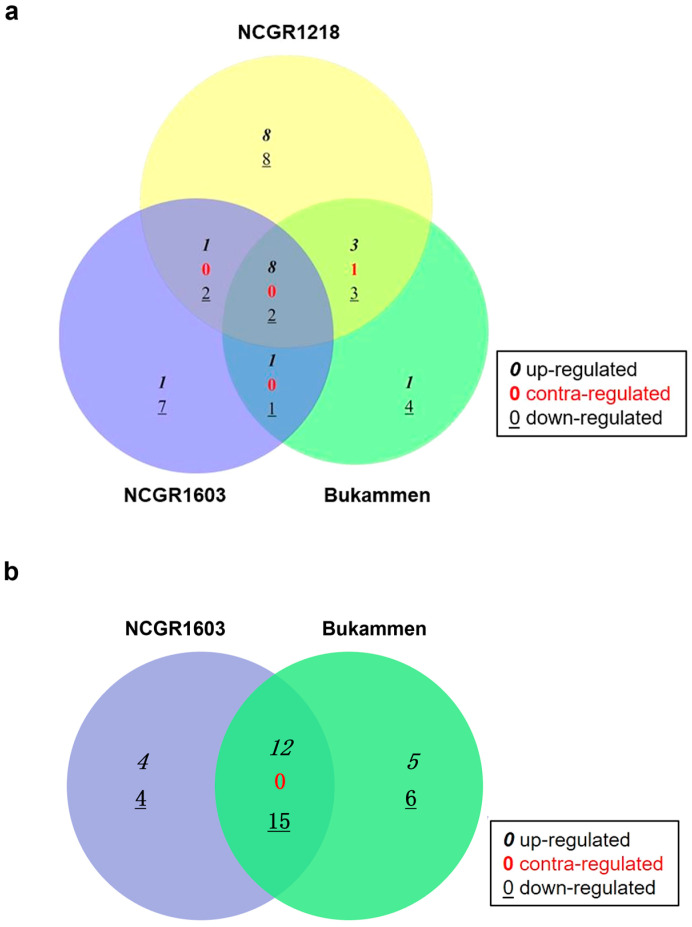
Number of shared differentially expressed genes (DEGs) in response to *Phytophthora cactorum* in the *RPc-1* locus of three *Fragaria vesca* genotypes. (**a**) The susceptible genotype NCGR1218, as well as the two resistant genotypes NCGR1603 and Bukammen, 48 h after wounding and inoculation with *P. cactorum* (their expression is relative to their untreated controls). The shared DEGs are listed in Appendix A. (**b**) Shared DEGs in the resistant genotypes NCGR1603 and Bukammen 48 h after wounding and inoculation with *P. cactorum* (the expression is relative to the susceptible genotype NCGR1218). DEGs were defined as fold change ≥2 or ≤−2 (*p* < 0.05). The DEGs shared by the resistant genotypes are listed in Appendix A.

**Figure 6 ijms-24-10851-f006:**
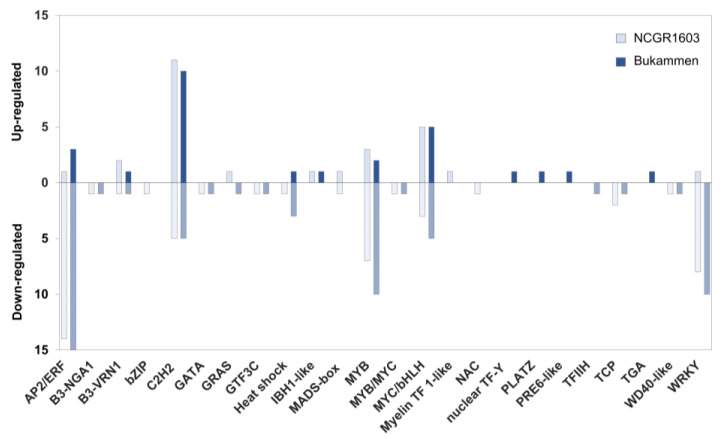
The number of upregulated and downregulated transcription factor genes in different TF families in the resistant *Fragaria vesca* genotypes NCGR1603 and Bukammen, 48 h after inoculation with *Phytophthora cactorum,* relative to the inoculated susceptible genotype NCGR1218. Genes with a fold change ≥2 or ≤−2 (*p* < 0.05) were defined as upregulated and downregulated, respectively.

**Figure 7 ijms-24-10851-f007:**
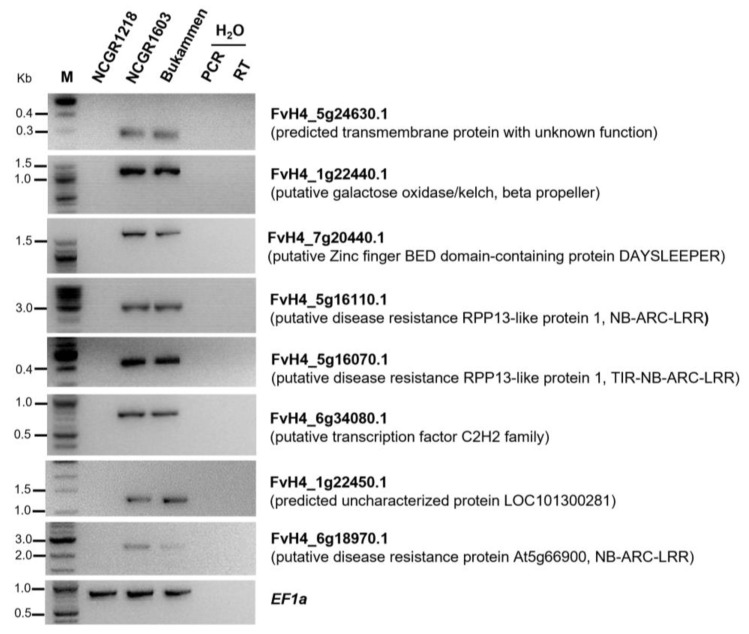
*Fragaria vesca* genes expressed in the two resistant genotypes NCGR1603 and Bukammen after wounding and inoculation with *Phytophthora cactorum*, while absent in the susceptible genotype NCGR1218. Agarose gel electrophoresis of PCR amplified transcripts generated by RT-PCR analysis. The RNA used for the cDNA synthesis was from a bulk of sixteen individual test plants from each genotype. The *F. vesca Elongation factor 1 alpha* (*EF1a*) gene was used as an internal control; H_2_O indicates template-free PCR and RT reactions, which were used as technical negative controls. M indicates a molecular marker. The original gel images were cropped and stacked for clarity and conciseness, and the full-length gel images are presented in Appendix A.

**Figure 8 ijms-24-10851-f008:**
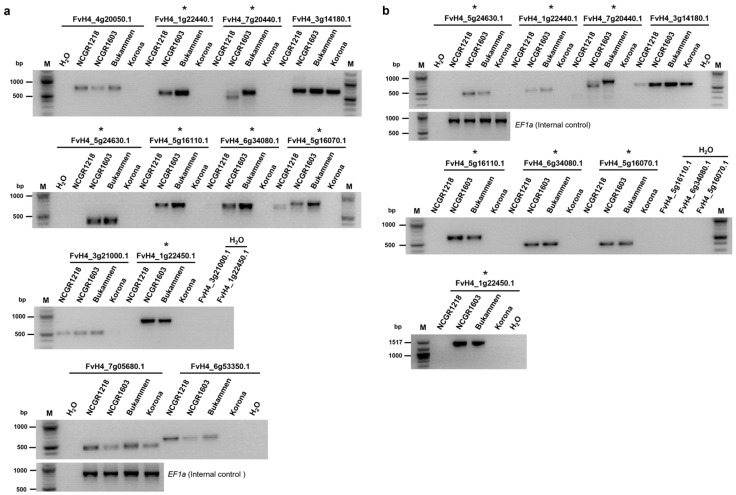
Gene presence–absence polymorphisms in strawberry genotypes that are resistant or susceptible to *Phytophthora cactorum*. The genes were selected based on lack of expression during infection of the susceptible genotype NCGR1218 (Appendix A). Gene fragments were amplified by PCR from genomic DNA, using up to two different primer pairs for each target gene. (**a**) Primer pair 1. (**b**) Primer pair 2 (see Appendix A). The genomic DNA used was from a bulk of ten individual test plants for each *Fragaria vesca* genotype or Korona, a moderately susceptible cultivar (*F. x ananassa*). Asterisks indicate genes detected only in the resistant *F. vesca* genotypes NCGR1603 and Bukammen. The *F. vesca Elongation factor 1 alpha* (*EF1a*) gene was used as an internal control; H_2_O indicates template-free PCR reactions (negative control). The original gel images were cropped for clarity and conciseness, and the full-length gel images are presented in Appendix A.

**Table 1 ijms-24-10851-t001:** Putative defence-related genes expressed only in the resistant *Fragaria vesca* genotypes NGCR1603 and Bukammen.

Gene ID	RNA Abundance 48 h after Inoculation (Mean Transcripts per Million ± SE)			Annotation Based on NCBI Non-Redundant Database and InterProScan Domain Analysis	Putative Role in Defence	Reference
	NCGR1218	NCGR1603	Bukammen			
FvH4_7g05680.1	0 ± 0.0	28 ± 1.0	33 ± 1.0	Protein DJ-1 homolog D-like	Cellular defence response against oxidative stress, and chloroplast development	[49,50]
FvH4_1g22440.1	0 ± 0.0	28 ± 1.0	35 ± 3	Putative galactose oxidase with kelch/beta-propeller	Production of H_2_O_2_ from galactose and O_2_; interaction with ASK1/MAP3K5	[51,52]
FvH4_6g53350.1	0 ± 0.0	27 ± 2.0	32 ± 2.0	Protease HtpX homolog, peptidase M48	Stress-controlled protease in *E. coli*; proteolytic quality control of misfolded proteins	[53,54]
FvH4_7g20440.1	0 ± 0.0	16 ± 1.0	19 ± 1.0	Zinc finger BED domain-containing protein DAYSLEEPER	Influence on global gene expression in *Arabidopsis thaliana*; BED-NLRs act as immune receptors in wheat (*Triticum aestivum*)	[55,56]
FvH4_5g03090.1	0 ± 0.0	11 ± 3.0	11 ± 3.0	Putative *Clostridium* epsilon toxin ETX/mosquitocidal toxin MTX2; agglutinin domain	Resistance to *Fusarium graminearum*	[57]
FvH4_3g14180.1	0 ± 0.0	10 ± 1.0	12 ± 1.0	Kinase RLK-Pelle-LRR-XI-1 family, protein kinase domain	Immune receptor against diverse pathogens; also involved in growth and development	[58,59]
FvH4_5g16110.1	0 ± 0.0	8 ± 0.5	8 ± 0.4	Putative disease resistance RPP13-like protein 1, leucine-rich repeat	Resistance to biotrophic oomycete *Peronospora parasitica*	[60]
FvH4_6g34080.1	0 ± 0.0	7 ± 0.4	7 ± 0.4	Putative transcription factor C2H2 family, zinc finger, RING-type	Response to abiotic stresses in plants	[61,62]
FvH4_5g32170.1	0 ± 0.0	6 ± 2.0	5 ± 1.0	Proteinase inhibitor PSI-1.2, proteinase inhibitor I20	Expression induced upon wounding and involved in plant defence against herbivory	[63,64]
FvH4_3g28530.1	0 ± 0.0	4 ± 0.5	3 ± 0.3	Glucan endo-1,3-beta-glucosidase 11-like	Response to pathogen infection in plants	[65,66]
FvH4_6g00620.1	0 ± 0.0	4 ± 0.2	5 ± 0.5	Putative transcription factor C2H2 family	Response to abiotic stresses in plants	[61,62]
FvH4_5g16070.1	0 ± 0.0	4 ± 0.2	3 ± 0.2	Putative disease resistance RPP13-like protein 1	Resistance to biotrophic oomycete *Peronospora parasitica*	[60]
FvH4_6g40960.1	0 ± 0.0	4 ± 0.2	3 ± 0.3	Cyclic nucleotide-gated ion channel 1-like	Active role in plant immunity in several pathosystems	[67,68]
FvH4_6g49500.1	0 ± 0.0	3 ± 0.8	3 ± 0.1	TMV resistance protein N-like, Toll/interleukin-1 receptor homology (TIR) domain	Resistance to *Tobacco mosaic virus* in *Nicotiana benthamiana*	[69]
FvH4_6g51870.1	0 ± 0.0	3 ± 0.4	2 ± 0.3	Putative transcription factor C2H2 family, zinc finger, RING-type	Response to abiotic stresses in plants	[61,62]
FvH4_3g45070.1	0 ± 0.0	3 ± 0.1	7 ± 1.0	Putative transcription factor C2H2 family, zinc finger, RING-type	Response to abiotic stresses in plants	[61,62]
FvH4_3g11010.1	0 ± 0.0	2 ± 0.2	4 ± 1.0	Berberine bridge enzyme-like 8, FAD linked oxidase, N-terminal	Enhanced resistance to *Botrytis cinerea*	[70]
FvH4_5g35070.1	0 ± 0.0	2 ± 0.4	4 ± 0.3	Putative pentatricopeptide repeat protein	Defence against biotic stress (*Pseudomonas syringae* pv. *tomato* and *Botrytis cinerea*) and abiotic stress	[71,72]
FvH4_5g11120.1	0 ± 0.0	2 ± 0.2	3 ± 0.1	Putative powdery mildew resistance protein, RPW8	Broad spectrum mildew resistance in *Arabidopsis thaliana*	[73]
FvH4_5g35080.1	0 ± 0.0	2 ± 0.5	4 ± 0.4	Putative pentatricopeptide repeat protein	Defence against biotic stress (*Pseudomonas syringae* pv. *tomato* and *Botrytis cinerea*)	[71,72]
FvH4_6g18970.1	0 ± 0.0	2 ± 0.3	2 ± 0.1	Probable disease resistance protein At5g66900	Unknown	_
FvH4_1g15330.1	0 ± 0.0	1 ± 0.2	1 ± 0.2	Probable disease resistance protein At5g66910	Unknown	_
FvH4_5g31070.1	0 ± 0.0	1 ± 0.3	2 ± 0.1	Putative pentatricopeptide repeat protein	Defence against biotic stress (*Pseudomonas syringae* pv. *tomato* and *Botrytis cinerea*) and abiotic stress	[71,72]

## Data Availability

All data sustaining the results in this study are included in this manuscript or its Appendix A. The datasets supporting the conclusions of this article are available in ArrayExpress under accession no. E-MTAB-12152, released on 1 January 2023. https://www.ebi.ac.uk/biostudies/arrayexpress/studies/E-MTAB-12152.

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
