# Peer review of "Comparative Transcriptome Analysis Reveals Novel Candidate Resistance Genes Involved in Defence against Phytophthora cactorum in Strawberry"

_ijms, 2023, doi:10.3390/ijms241310851_

Round 1

Reviewer 1 Report (New Reviewer)

Authors submitted a manuscript entitled “Comparative Transcriptome Analysis Reveals Novel Candidate Resistance Genes Involved in Defence Against Phytophthora cactorum in Strawberry” to IJMS. In this study, they focused on crown rot, a damaging disease in strawberries caused by Phytophthora cactorum. While most commercial strawberry cultivars are susceptible, the diploid species Fragaria vesca holds the potential for resistance genes. By analyzing the transcriptome profiles of two resistant F. vesca genotypes and one susceptible genotype, authors identified several defense-related genes expressed at higher levels in the resistant genotypes. Additionally, they found specific genes associated with a previously identified resistant gene locus. The study highlights potential candidate genes for further investigation into plant defense mechanisms against P. cactorum. The manuscripts is written well but few sections needs improvements.

Abstract
L23-24; Have you analyzed the genome sequence to compare these genes between contrasting genotypes?

L24-27; These sentences are unclear? Authors found RPc-1, previously, but in next sentence it mentioned that 4-5% genes are differentially expressed in RPc-1. What does it mean?

The abstract is not written very well.

The discussion lacks statements about key findings, gaps, and future research directions.

The conclusion also needs to be improved.

Overall English is fine but still needs careful checking of sentences and grammatical mistakes.

Author Response

Replies to reviewer 1  

  1. Abstract
    L23-24; Have you analyzed the genome sequence to compare these genes between contrasting genotypes?

Response: We have not analysed the genome sequences for these genotypes (NCGR1218, Bukammen and NCGR1603) as they are not available in the public databases (e.g., GDR or NCBI databases). We tested the presence or absence of these genes in the genomic DNA of these genotypes by PCR using two primer pairs as described in the materials and methods (lines600-602) and in the figure legend (Figure 8) in lines 379-380, in the revised manuscript. To make this clear in the ABSTRACT, we have added a few words in line 24 in the revised manuscript, which now reads as follows “Seven of these R-genes were expressed only in the resistant genotypes and not in the susceptible genotype, and these appeared to be present only in the genomes of the resistant genotypes, as confirmed by PCR analysis” (lines 22-24 in the revised manuscript).

  1. L24-27; These sentences are unclear? Authors found RPc-1, previously, but in next sentence it mentioned that 4-5% genes are differentially expressed in RPc-1. What does it mean?

Response: These sentences means that out of total 800 genes present in the RPc-1 locus, only 4-5% of the genes (35-38 out of total 800 genes) in the RPc-1 were differentially expressed in the resistant genotypes compared to the susceptible genotype after inoculation with P. cactorum. To make this clear in the ABSTRACT, we have revised the sentence to make it clearer (lines 26-28 in the revised manuscript). 

  1. The abstract is not written very well.

Response: To improve the abstract, we have revised line 24 and lines 26-28 (in response to comment 1 and 2 above). We have also deleted the sentence on genes involved in flavonoid and isoprenoid biosynthesis, besides phytohormone signalling pathways (lines 31-33 in the revised manuscript) to make the abstract more concise (please also see the response to reviewer 3; comment 1).

  1. The discussion lacks statements about key findings, gaps, and future research directions.

Response: To improve the discussion and to highlight key findings, we have added a sentence in lines 394-397399-403, and revised a sentence in lines 447-448. Furthermore, we have added information about gaps and future research directions in lines 402-403, lines 470-471 and lines 506-508 in the revised manuscript. 

  1. The conclusion also needs to be improved.

Response: To improve the conclusions section, we have added information about the candidate resistance genes identified in the present study and a sentence highlighting the future prospects of our work (lines 608-613 in the revised manuscript). 

General comments:

Does the introduction provide sufficient background and include all relevant references? -Can be improved

Response: To improve the introduction, we have added two sentences and new references on the use of fungicides in controlling crown rot and leather rot diseases in strawberry, and the current challenges for their use in strawberry farming (lines 47-53 in the revised manuscript). We have also added a citation to a completely new review by Chen et al. 2023 in line 40 of the revised manuscript. Furthermore, we have added a few words to emphasize the involvement of the root system during P. cactorum infection (line 44 of the revised manuscript).

Reviewer 2 Report (New Reviewer)

This study holds significance as it utilized Comparative Transcriptome Analysis to identify potential disease resistance gene candidates for P. cactorum infection. The well-planned design and executed experiments provide valuable information. However, I would like to offer the following comments:

1. The rationale behind selecting a harvest time of 48 hours after inoculation as the most appropriate is unclear. This is particularly relevant considering that necrotic areas appeared in the susceptible genotype NCGR1218 after 2 weeks, while no symptoms were observed in the resistant genotypes NCGR1603 and Bukammen until 4 weeks later. Moreover, the majority of putative R-genes (26 out of 31 highly expressed genes in the resistant genotypes) were constitutively expressed, regardless of pathogen inoculation. Additionally, the resistant line did not exhibit a significant increase in flavonoid and isoprenoid compounds. Further clarification is necessary to establish the optimal sampling period (after 48 hours) for the study.

2. The criteria used to measure disease scores after inoculation with Phytophthora cactorum in Figure 1 should be clearly described. How was the degree of wilting and necrosis determined in plants? Was it based on visual observations of factors such as color or other observable indicators? Are there any additional measurable criteria besides visually determined scores as described in figure legend?

3. Figure 2 could be presented in a simpler manner by using different symbols instead of complex expressions like N12C00, N16I48, BukW48, etc.

4. The text in the "Abstract" and other sections should be presented in its final manuscript form. Some changes are currently displayed as they are, so they need to be modified accordingly.

5. There are duplicate expressions in the manuscript, specifically in lines 22-24. Corrections are needed to avoid repetition.

Author Response

Replies to reviewer 2

This study holds significance as it utilized Comparative Transcriptome Analysis to identify potential disease resistance gene candidates for P. cactorum infection. The well-planned design and executed experiments provide valuable information. However, I would like to offer the following comments:

  1. The rationale behind selecting a harvest time of 48 hours after inoculation as the most appropriate is unclear. This is particularly relevant considering that necrotic areas appeared in the susceptible genotype NCGR1218 after 2 weeks, while no symptoms were observed in the resistant genotypes NCGR1603 and Bukammen until 4 weeks later. Moreover, the majority of putative R-genes (26 out of 31 highly expressed genes in the resistant genotypes) were constitutively expressed, regardless of pathogen inoculation. Additionally, the resistant line did not exhibit a significant increase in flavonoid and isoprenoid compounds. Further clarification is necessary to establish the optimal sampling period (after 48 hours) for the study.

Response: We used 48 hours after inoculation for sampling based on a temporal expression study of defense related genes in F. vesca genotypes in response to P. cactorum that we previously reported (Chen et al. 2016). Another research group reported that visible hyphae was observed on the strawberry root surface 48 hours after inoculation, representing the early infection stage of P. cactorum (Toljamo et al. 2016). An explanation for this sampling time point was described in the previous version of the manuscript in the materials and methods (lines 520-524 in the revised manuscript). Wilting and collapsing of the whole plant was observed in the genotype NCGR1218 from the first week onwards and most plants died within two weeks after inoculation with P. cactorum (Figure 1b), which means that infection likely started before the first week passed. We would like to point out that intense transcriptome reprogramming was observed in all the tested F. vesca genotypes 48 hours after inoculation with P. cactorum as shown in the multidimensional scaling plot (Figure 2) of the revised manuscript. This signifies that 48 hours after inoculation is at least a relevant time point to examine. The constitutive expression of the majority of the R-genes in the resistant genotypes, irrespective of pathogen inoculation, is possibly due to constant activation of defence, as observed by others in a resistant genotype of F. vesca attacked by Botrytis cinerea (Zhao et al. 2022). This was in fact discussed in the previous version of the manuscript (lines 395-397 in the revised manuscript). It is true that the resistant genotypes did not exhibit a significant increase in expression of genes linked to flavonoid and isoprenoid compounds, as well as phytohormones signalling pathways compared to the susceptible genotype after inoculation. An explanation why significant difference was not observed in the resistant genotypes for genes in these pathways was discussed in the previous version of the manuscript (lines 488-491492-495 in the revised manuscript).

To make selection of the time point 48 h clearer, we have added a few words in the materials and methods section in the revised manuscript (lines 520-524).

  1. The criteria used to measure disease scores after inoculation with Phytophthora cactorumin Figure 1 should be clearly described. How was the degree of wilting and necrosis determined in plants? Was it based on visual observations of factors such as color or other observable indicators? Are there any additional measurable criteria besides visually determined scores as described in figure legend?

Response: The disease scoring was performed based on visual symptom observations as originally described by Bell et al. 1997 and later by Eikemo et al. 2010. No additional measurable criteria was used besides visual symptom observations. To make this clearer we have added a few words including citations to previous publications (Bell et al. 1997 and Eikemo et al. 2010) that we base our scoring system on, to the legend of Figure 1 (lines 131-137).

  1. Figure 2 could be presented in a simpler manner by using different symbols instead of complex expressions like N12C00, N16I48, BukW48, etc.

Response: We appreciate your suggestion to improve the clarity of Figure 2 by replacing complex expressions like N12C00, N16I48, and BukW48 etc. used in the original manuscript. We have replaced the complex expressions with coloured symbols in Figure 2 in the revised manuscript. As a consequence of this we have revised the figure legend (lines 163-168). We have also replaced the part of the legend title of Figure 2 from “principal coordinate analysis” to “multidimensional scaling plot”, which we learned is more appropriate (line 155 in the revised manuscript).

  1. The text in the "Abstract" and other sections should be presented in its final manuscript form. Some changes are currently displayed as they are, so they need to be modified accordingly.

Response: We have now removed the previous track changes from the Abstract and other sections. Only new track changes made in response to the current reviewer comments are displayed in the revised manuscript. 

  1. There are duplicate expressions in the manuscript, specifically in lines 22-24. Corrections are needed to avoid repetition.

Response: The first part of the sentence is about expression of R genes while the second part is about the presence of these genes in the genomes. To avoid possible misinterpretation, we have added a few words in line 24 in the revised manuscript (see also response to reviewer 1; comment 1).

Reviewer 3 Report (New Reviewer)

Please shorten the abstract according to the rules od the publisher and focus more on the main findings of the manuscript.

In general the manuscript is too long and should be shortened by moving some figures to the supplement.

Methods 4.5. While RT-PCR with semi-quantitative analysis is useful, quantitative approach would be preferred as giving higher precision (this is comment for future studies).

In the Conclusions section include future prospects and how this data can be actionable in the aspect of the strawberry resistance.

As a reviewer, I appreciate attaching the original photos of the electrophoresis gels.  

Author Response

Replies to reviewer 3

  1. Please shorten the abstract according to the rules od the publisher and focus more on the main findings of the manuscript.

Response: We have deleted the sentence on genes involved in flavonoid and isoprenoid biosynthesis, as well as phytohormone signalling pathways from the abstract to make it more concise (lines 31-33). A few sentences were added to the abstract after suggestions from previous reviewers and the editor. These include our previous work on the F. vesca genotypes (lines 13-14) and the RPc-1 locus (lines 24-26) in the revised manuscript.

  1. In general, the manuscript is too long and should be shortened by moving some figures to the supplement.

Response: We appreciate and thank you for the suggestion. Thus, we have moved Figure 3 (Gene Ontology (GO) analysis of the differentially expressed genes (DEGs) in the resistant Fragaria vesca genotypes) from the main text to the supplementary. This figure is Supplementary Figure S1 in the revised manuscript. Consequently, all other figures, both main and supplementary, have changed numbers in the revised manuscript. Figure 8 was originally placed as supplementary, but during previous revisions, it was suggested to move it from supplementary to the main manuscript. Hence, we would like to keep Figure 8 in the main manuscript as it is now. 

  1. Methods 4.5. While RT-PCR with semi-quantitative analysis is useful, quantitative approach would be preferred as giving higher precision (this is comment for future studies).

Response: We appreciate and thank you for the valuable suggestion. For future studies, we will aim for a quantitative approach (e.g., qPCR) instead of semi-quantitative RT-PCR.

  1. In the Conclusions section include future prospects and how this data can be actionable in the aspect of the strawberry resistance.

Response: To improve the conclusions section, we have added information about the candidate resistance genes identified in the present study (lines 608-612 in the revised manuscript). We have also added a sentence highlighting the future prospects of our work (lines 612-613 in the revised manuscript). 

This manuscript is a resubmission of an earlier submission. The following is a list of the peer review reports and author responses from that submission.

Round 1

Reviewer 1 Report

·         This study is intended to reveals genes involved in defence against phytophthora cactorum in Strawberry by comparative transcriptome analysis. However, the content is too superficial to explain this problem only through disease stress transcriptome analysis and analysis of each type of gene. Biological experiments are needed to locate and clone the genes of target traits for functional analysis. At present, the innovation is not enough, the content is relatively simple, not enough to publish.

Author Response

see enclosed file

Reviewer 2 Report

This article is still very detailed overall, especially the expression verification of the genes screened by the transcriptome.  The author has done a lot of work on this. However, the disease resistance process of plants is complex and rapid. I have the following questions to communicate with the author.

1) What is the research progress on Phytophthora cactorum in Strawberry? Has any related disease resistance gene been found in strawberry?

2) Why did the author choose only one time point (48h) for the study?  As far as I know, the sampling time of many transcriptome articles studying the process of plant disease resistance has a sequence.

3) Plant disease resistance is a very complicated process, and the expression of many genes does not always show an upward trend in the process of disease resistance.

4) I have also been engaged in the identification of disease resistance genes.  The analysis of gene expression is only a preliminary criterion, and subsequent identification work is still needed.

Author Response

see enclosed file

Reviewer 3 Report

line no 20 the numbering on abstract is confusing . does number 1 refers to the first finding? If that is the case how about number 2?
line no 136 again the numbers are confusing, what does 4- , 3- , 2- refers to ? actual count or number of sample?
Figure : can author provide hierarchical clusetring of genes of various categories such as virulence, defense mechanisms, R genes etc and provide the heatmap. Figure 1
can be modified

Also author can point out some pathways regulated?
 can author provide the gene enrichment analysis ( Go terms) for the DEG's

Author Response

see enclosed file

Round 2

Reviewer 1 Report

The author has done a lot of work on the expression of the genes screened by the transcriptome, BUT it was not  enough to verify the genes. I think it is not appropriate to publish in IJMS.

Reviewer 2 Report

The author basically responded to my comments completely, although regarding the last question, I am somewhat disappointed that the author responded that the question was beyond the scope of the current research.

But based on the author‘s detailed response, and the future prospects of this research, I agree with the revision of the article.